# Targeted Proteomic Analysis of Patients with Ascending Thoracic Aortic Aneurysm

**DOI:** 10.3390/biomedicines11051273

**Published:** 2023-04-25

**Authors:** Aphrodite Daskalopoulou, Sotiria G. Giotaki, Konstantina Toli, Angeliki Minia, Vaia Pliaka, Leonidas G. Alexopoulos, Gerasimos Deftereos, Konstantinos Iliodromitis, Dimitrios Dimitroulis, Gerasimos Siasos, Christos Verikokos, Dimitrios Iliopoulos

**Affiliations:** 1Laboratory for Experimental Surgery and Surgical Research “N.S. Christeas”, Athens Medical School, National and Kapodistrian University of Athens, 115 27 Athens, Greece; 2Second Department of Cardiology, National and Kapodistrian University of Athens, 115 27 Athens, Greece; 3Cardiology Department, General Hospital of Chalkida, 341 00 Chalkida, Greece; 4Protatonce Ltd., Demokritos Science Park, 153 43 Athens, Greecevicky.pliaka@protavio.com (V.P.); leo@protatonce.com (L.G.A.); 5Department of Mechanical Engineering, National Technical University of Athens, 106 82 Athens, Greece; 6Department of Cardiology, G. Gennimatas, General Hospital of Athens, 115 27 Athens, Greece; 7School of Medicine, Witten/Herdecke University, 58455 Witten, Germany; 8Second Department of Propedeutic Surgery, Laiko General Hospital, School of Medicine, National and Kapodistrian University of Athens, 115 27 Athens, Greece; 9Third Department of Cardiology, National and Kapodistrian University of Athens, 115 27 Athens, Greece

**Keywords:** ascending aortic thoracic aneurysm, proteomics, biomarkers, C-C motif chemokine ligand 5, defensin beta 1, intracellular adhesion molecule-1

## Abstract

Background: There is a need for clinical markers to aid in the detection of individuals at risk of harboring an ascending thoracic aneurysm (ATAA) or developing one in the future. Objectives: To our knowledge, ATAA remains without a specific biomarker. This study aims to identify potential biomarkers for ATAA using targeted proteomic analysis. Methods: In this study, 52 patients were divided into three groups depending on their ascending aorta diameter: 4.0–4.5 cm (*N* = 23), 4.6–5.0 cm (*N* = 20), and >5.0 cm (*N* = 9). A total of 30 controls were in-house populations ethnically matched to cases without known or visible ATAA-related symptoms and with no ATAA familial history. Before the debut of our study, all patients provided medical history and underwent physical examination. Diagnosis was confirmed by echocardiography and angio-computed tomography (CT) scans. Targeted-proteomic analysis was conducted to identify possible biomarkers for the diagnosis of ATAA. Results: A Kruskal–Wallis test revealed that C-C motif chemokine ligand 5 (CCL5), defensin beta 1 (HBD1), intracellular adhesion molecule-1 (ICAM1), interleukin-8 (IL8), tumor necrosis factor alpha (TNFα) and transforming growth factor-beta 1 (TGFB1) expressions are significantly increased in ATAA patients in comparison to control subjects with physiological aorta diameter (*p* < 0.0001). The receiver-operating characteristic analysis showed that the area under the curve values for CCL5 (0.84), HBD1 (0.83) and ICAM1 (0.83) were superior to that of the other analyzed proteins. Conclusions: CCL5, HBD1 and ICAM1 are very promising biomarkers with satisfying sensitivity and specificity that could be helpful in stratifying risk for the development of ATAA. These biomarkers may assist in the diagnosis and follow-up of patients at risk of developing ATAA. This retrospective study is very encouraging; however, further in-depth studies may be worthwhile to investigate the role of these biomarkers in the pathogenesis of ATAA.

## 1. Introduction

Thoracic aortic aneurysms (TAA) often go undetected, to the point at which they occur brutally, causing terrible consequences [1]. Aortic dissection and rupture account for 2–7% of all sudden cardiac fatalities in the general population [2,3], which represents a significant mortality load. Up to 22% of patients who experience acute aortic events are pronounced dead before reaching the hospital [4]. It is estimated that in the Unites States alone, aortic diseases account for about 13.000 deaths annually and act as a contributing factor in more than 16.415 deaths [5], making them the 17th most common cause of mortality in people older than 50 years of age [6]. Moreover, many thoracic aortic-related sudden deaths can be misdiagnosed as myocardial infarction. As such, it is possible that these data significantly underestimate the true burden of aortic diseases in the population. This evidence highlights the significance of timely detection of TAA, as it is rampant but asymptomatic and hence termed a “silent killer” [1].

TAA can be categorized as either heritable or degenerative from an etiologic perspective. Less than 30% of all TAA cases are genetically triggered, whereas more than 70% are degenerative [7]. Mutations in genes encoding proteins such as smooth muscle (SM) contractile proteins, extracellular matrix (ECM) proteins and proteins involved in transforming growth factor beta (TGF-β) signaling are the main causes of genetically triggered TAA [8]. Sporadic TAA are primarily linked to risk factors such as age, male sex, smoking and hypertension [9,10]. In the context of aneurysmal diseases of the thoracic aorta, ascending (ATAA) and descending (DTAA) aneurysms behave as two distinct types of disorders. This could be explained by the different embryologic origins of ascending and descending aortic vascular smooth muscle cells (VSMCs), which are in charge of secreting many of the proteolytic factors associated with aneurysm formation, including matrix metalloproteinase (MMP) and plasmin [11,12].

Currently, the diagnosis of ATAA is largely based on imaging tests (echocardiography, computed tomography and magnetic resonance imaging), which are frequently performed for unrelated purposes; therefore, the aneurysm is found by accident [13]. However, ATAA in patients who are not subjected to these imaging studies remain undetected, and complications such as aortic dissection and rupture may occur. There are no effective preventive strategies for TAA; thus, early detection, surveillance and treatment are critical to improving outcomes.

The development of biomarkers that could be helpful in identifying individuals with thoracic aortic illness has been investigated [14,15]. However, to date, the majority of biomarkers are mostly successful in identifying aortic disease after an aortic dissection or rupture has taken place. Hence, in order to improve clinical care and outcomes for this fatal disease, it is essential to establish new ways to help in the identification of people at risk of currently having or developing a thoracic aneurysm. The 12 proteins selected for measurement were carefully chosen based on their known involvement in several cellular pathways linked to ATAA, including inflammation, immune cell activation, extracellular matrix degradation, smooth muscle cell proliferation and differentiation, and endothelial dysfunction. Using a targeted proteomic approach, we sought to investigate the effect of the formation of an ascending thoracic aortic aneurysm on the proteomic profile in the serum of patients identified with ATAA, in an effort to develop potential biomarkers for the detection of ATAA.

## 2. Materials and Methods

### 2.1. Patient Population

This study was conducted at the General Hospital of Chalkida, a tertiary care center in Greece, and the samples were obtained consecutively from May 2018 until January 2020. The approval code for our study is 56; 13 October 2017, granted by the Ethics Committee of the General Hospital of Chalkida, the Medical Center Karistos, and the Medical Center Kymi. Some 52 patients undergoing degenerative ATAA were included in the study population, as shown in the flowchart for enrollment in Figure 1. Based on the current practice guidelines, in accordance with recent analyses [1,16,17], patients were categorized into three groups depending on aorta diameter: 4.0–4.5 cm (*N* = 23), 4.6–5.0 cm (*N* = 20), >5.0 cm (*N* = 9). All subjects were clinically evaluated to exclude hereditary forms. A total of 30 controls were in-house populations ethnically matched to cases without known or visible TAA-related symptoms and with no TAA familial history. Before the beginning of our study, all patients provided medical history and underwent physical examination (Table 1). Diagnosis was assessed by echocardiography and confirmed by angio-computed tomography (CT) scans.

Clinical information was obtained from the patients’ medical records prior to their index echocardiographic examination, including history of hypertension and other cardiovascular risk factors, as well as conditions listed in the exclusion criteria below. These were identified using International Classification Disease (ICD) 11 diagnosis codes. History of smoking was defined as current or former smokers based on patient-provided information. Demographic data including age, sex, height, weight, blood pressure and heart rate measurements were obtained upon index echocardiographic examination. Body surface area (BSA) was calculated according to the Du Bois formula.

The exclusion criteria included the presence of acute inflammation, infection or malignancy, as well as patients with TAA attributed to genetic syndromes such as Marfan Ehlers–Danlos, Loeys–Dietz and Turners, patients with a bicuspid aortic valve, patients with aortitis due to infectious (syphilis, salmonella, staphylococcus, mycobacterium) or inflammatory causes (giant cell and Takaysu’s arteritis, Behcet’s disease, Cogan syndrome, polychondritis, rheumatoid arthritis, vertebral arthropathy). Patients with post-traumatic aneurysms (casts, pseudo-aneurysms, chronic dissections) were also excluded from this study. This study was conducted in agreement with the principles outlined in the Declaration of Helsinki. Written informed consent was obtained from all subjects according to the research protocol approved by the Ethical Review Board of the General Hospital of Chalkida.

### 2.2. Definition of ATAA

ATAA is defined as an ascending thoracic aorta with a diameter ≥ 4.0 cm [18,19]. Given that studies have used different definitions of aneurysm, a threshold of 4.0 cm was chosen to maximize the sensitivity of finding an expanded aorta in the screening context. Diagnosis was assessed by echocardiography and confirmed by CT scans.

Measurements of the mid ascending aorta diameter were performed in a standardized manner across the hospital using a typical ultrasound equipment (Vivid 7 PRO, GE Healthcare, Chicago, IL, USA) (Figure 2). Experienced cardiologists/echocardiographers performed transthoracic echocardiograms (TTE) in the left lateral decubitus position. Cardiologists typically image the mid ascending level of the tubular ascending aorta above the proximal segment by scanning one to two intercostal spaces above the usual parasternal long axis view; in such views, the aortic valve is typically no longer visualized. Following the recommendations of the American Society of Echocardiography (ASE), diameters were measured perpendicular to the long axis of the aorta using the leading edge to leading edge approach of the maximum distance between the anterior and posterior aortic walls at end diastole [20]. It is important to note that at least three cardiac cycles were obtained.

The aortic sinus of Valsalva, the sinotubular junction (STJ), 1 cm above the STJ and the maximum size of the ascending aorta were the four levels at which the off-line measurements of the ascending aorta were conducted during the TTE. The biggest diameter and the maximum detectable length of the ascending aorta above the ST junction were both measured at the sinus of Valsalva level. The inner wall to inner wall convention was used to measure aortic diameters during diastole, with multiple cycles carried out at each level as necessary to increase accuracy. We deemed the inner wall to differ from the inner edge convention by moving the cursor to include the inner portion or the wall beyond the inner edge. Diastolic measures were identified by the onset of the QRS on the electrocardiogram (ECG), or if an adequate ECG was not available, by the closure of the aortic valve and a downward motion of the aortic wall.

ATAA was confirmed by CT scans by the double oblique technique measurement, in which the cross-sectional aorta was determined by averaging the outer-to-outer wall diameters measured at an angle of 60° from one another. Between the sinotubular junction and the proximal take-off of the innominate artery, the greatest part of the aorta was measured. A senior member of the cardiology faculty with expertise in cardiac and thoracic imaging performed the measurements using Syngo.via VB30 imaging software (Siemens Healthineers, Erlangen, Germany). Since the current guideline recommendations for surgical intervention are based on the diameter without indexing, we did not index the aorta diameter by body size [16]. We only took into account aneurysms in the ascending aorta, because the aortic root measurement is prone to motion abnormalities throughout the cardiac cycle, and the scans included both cardiac-gated and non-gated scans.

### 2.3. Blood Sampling

Blood samples were collected under fasting conditions by venipuncture; these samples were used for biochemical measurements (Table 2). Low-density lipoprotein (LDL) cholesterol was calculated according to the Friedewald’s equation. The serum samples were separated by centrifugation within 30 min at 3300× *g* for 20 min and subsequently stored at −80 °C until analysis. Prior to running dual-antibody Luminex assays, serum samples were heat-inactivated to eliminate complements. The heat-inactivation was performed as follows. After gentle mixing, the samples were slowly thawed at room temperature. They were then heated at 56 °C for 30 min in Bain–Marie and gently mixed. Inactivated serum samples were used for the measurements.

### 2.4. Proteomics

Utilizing the multiplex assay service provided by ProtATonce (Athens, Greece), we developed 12 custom dual-antibody Luminex assays of potential biomarkers. Between 2 and 5 antibodies were chosen and purified from carrier proteins and buffers containing amines that interfere with the coupling and biotinylation processes. All antibodies were tested in pairs, as capture and as detection antibodies. Capture antibodies were linked to the magnetic beads, while detection antibodies were biotinylated. The efficiency of biotinylation and coupling was validated by quality control. The best capture/detection antibody combination for each biomarker was chosen using signal-to-noise ratio measurements. The secondary antibody concentration was assessed based on its signal and its noise (off-target signal) in the bead panel. The 12-plex panel’s cross reactivity was determined by assaying the 12-plex bead mix against each single detection antibody in the presence of a 12-plex recombinant protein mix. Based on the European Medicines Agency’s EMEA/CHMP/EWP/192217/2009 guidelines on validating bioanalytical methods, assay validation, including limit of detection (LOD) and reproducibility, was carried out.

### 2.5. Luminex Assay Principle

For the preparation of the bead mix and the detection mix, 12 capture antibodies coupled to Luminex magnetic beads and 12 biotinylated detection antibodies were multiplexed, respectively. Each sample is incubated with the bead mix in a well of a 96-well microliter plate to allow binding of the analyte. Any unbound material was removed by washing using a magnetic separator. The formed antibody–analyte complex is incubated with the detection mix, which is also specific for each analyte. Any unbound detection antibody is removed by a washing step and the formed complex of antibody–analyte–detection antibody is labelled with streptavidin R-phycoerythrin (SAPE, Cat Nr: S866, Invitrogen, Carlsbad, CA, USA). The fluorescent emission of R-phycoerythrin and the distinct microsphere fluorescent signatures are measured simultaneously by the Luminex^®^ xMAP™-compatible analyzers. A mixture of reference standard proteins is also measured to generate a calibration curve that is used to calculate the absolute concentrations of each analyte in the sample.

The following 12 proteins were measured in patients’ serum; interleukin-1α (IL-1α), IL-8, C-C motif chemokine ligand 5 (CCL5), C-X-C motif chemokine 11 (CXCL11), follistatin (FST), intracellular adhesion molecule-1 (ICAM-1), prokineticin 1(PROK1), resistin (RETN), matrix metallopeptidase 9 (MMP-9), tumor necrosis factor alpha (TNFα), transforming growth factor-beta 1 (TGFB1), and defensin beta 1 (HBD1).

### 2.6. Statistical Analysis

Mean fluorescence intensity (MFI) values were used as input of all the downstream analyses. Three or more parameters were compared using a Kruskal–Wallis test. Analysis of selective data pairs using the Dunn’s multiple comparison test was conducted. All t-tests were two-tailed, and a nonparametric distribution was assumed. The Mann–Whitney test for unpaired data was used to assess differences between medians. The overall diagnostic performance was summarized using non parametric area under the ROC curve (AUC). *p*-values were considered significant if *p* < 0.05.

## 3. Results

### 3.1. Study Population

After applying inclusion and exclusion criteria, 52 patients (57.7% male) with ATAA (mean age: 71.27 ± 11.56) and 30 controls (56.7% male, mean age: 71.97 ± 11.28) were eligible for analysis. Clinical and echocardiographic characteristics are tabulated in Table 1. All patients were diagnosed with ATAA with an average ascending aorta diameter of 4.65 ± 0.32 cm. The prevalence of cardiovascular risk factors (hypertension, hyperlipidemia, angina) was similar between controls and patients. Non-proteomic blood measurements are shown in Table 2. In order to avoid the need for propensity score matching to correct for confounding factors, and to evaluate the impact of the aorta diameter per se on the proteome, similar baseline characteristics of the patients in each group were chosen as recruitment criteria. As a result, a comparison of the demographic data for the two groups (gender, age, BSA, BMI, glycemic and lipidemic profile; *p* > 0.05) revealed no differences between them.

### 3.2. Evaluation of D-Dimer as a Potential Diagnostic Biomarker for ATAA

To begin the screening of the potential differentially distributed serum markers among patients with ATAA and healthy controls, peripheral blood samples were collected at the time of admission, and laboratory tests were performed to investigate the levels of a specific marker for venous thromboembolism, D-dimer. D-dimer expression was elevated in the general patient population, with a mean of 1.01 ± 1.09 mg/L (*p* < 0.05). However, when we examined the D-dimer levels of the individual pathological groups, we observed that only the group with aortic diameter range 4.6–5.0 cm had significantly higher D-dimer expression in comparison to the control group (*p* < 0.05, Figure 3A). The absolute value of D-dimer cannot fully reflect the health status of patients, since whether D-dimer is in the normal range is more clinically relevant. To better visualize the potential predictive efficacy of D-dimer for AΤAA, the 52 patients were divided into two groups based on D-dimer levels: D-dimer ≤ 0.5 mg/L and D-dimer > 0.5 mg/L, which correspond to the normal and pathological range, respectively. As shown in Figure 3C, for 46.2% of ATAA patients, D-dimer levels were >0.5 mg/L, while for 53.8% of ATAA patients, D-dimer expression was ≤0.5 mg/L. In the control group, 30% of the population had D-dimer levels > 0.5 mg/L. This suggests that D-dimer cannot be considered a reliable biomarker for the diagnosis of ATAA.

### 3.3. Targeted Proteomic Analysis for the Investigation of Potential Diagnostic Biomarkers for ATAA 

Out of the 12 proteins included in the proteomic study, 6 were significantly expressed in the serum of patients with ATAA. A Kruskal–Wallis test revealed a potent induction of CCL5, HBD1, ICAM1, IL8 and TNFα expression in patients with ATAA in comparison to control subjects (*p* < 0.0001) (Figure 4A–E). In particular, when a Dunn’s multiple comparison test was performed, we observed significantly high levels of CCL5, HBD1, ICAM1 and IL8 (*p* < 0.0001) in patients with aortic diameter range 4.0–5.0 cm in comparison to the control subjects with a normal aortic diameter. This expression remained high (*p* < 0.01) in patients whose aorta was >5.0 cm. TNFα levels were significantly elevated in ATAA patients compared to controls, but the calculated significance differed among the three aortic diameter groups (*p* = 0.0366, *p* = 0.0002 and *p* = 0.0055, respectively). Regarding the growth factor TGFB1, significantly high levels were found (*p* = 0.0013) in ATAA patients (Figure 4F).

### 3.4. Diagnostic Performance of the Identified Proteins

To further evaluate the diagnostic efficacy of the mentioned markers, the receiver operating characteristic (ROC) curve of D-dimer and the 6 studied proteins mentioned above was performed. The area under the curve (AUC) found on receiver-operating characteristics curve analysis for all 52 ATAA patients versus all control subjects was 0.64 (95% CI, 0.52 to 0.76), 0.84 (95% CI, 0.76 to 0. 93), 0.83 (95% CI, 0.73 to 0.93), 0.83 (95% CI, 0.73 to 0.92), 0.70 (95% CI, 0.58 to 0.83), 0.69 (95% CI, 0.56 to 0.81) and 0.76 (95% CI, 0.65 to 0.87) for D-dimer, CCL5, ICAM1, HBD1, IL8, TNFα and TGFB1, respectively (Figure 5A–G). Figure 5H summarizes this detailed information on ROC curves. D-dimer showed the least favorable overall diagnostic performance compared with the studied markers (Figure 5H). These results indicate that CCL5, ICAM1 and HBD1 showed the most encouraging diagnostic performance for the diagnosis of ATAA.

### 3.5. Targeted Proteomic Analysis to Investigate the Specificity of the Potential Biomarkers for the ATAA

To further establish the specificity of the investigated biomarkers for the ATAA, we added a subgroup to our studies. A total of 15 patients with history of cardiovascular disease (CVD) (coronary artery disease, arrhythmias, hypertension and cardiomyopathy) and a physiological aortic diameter (mean ascending aorta diameter: 3.40 ± 0.33 cm) were enrolled in our study, and their proteomic profiles were examined. Demographic characteristics and medical history were recorded. In order to avoid the presence of confounding factors, identical baseline characteristics of the CVD patients with the previously researched groups were utilized as recruiting criteria for legibility. Thus, CVD participants were matched for gender, age, BMI, BSA, lipidemic and glycemic profile with the control subjects, as well as the patients with ATAA.

As indicated in Figure 6, out of the 6 proteins that were found to be significantly expressed in the serum of patients with ATAA in comparison to the control subjects, 5 were also significant for ATAA compared to the CVD patients. In particular, a one-way ANOVA on ranks showed that CCL5, HBD1, ICAM1, IL8 and TNFα levels were increased in patients with ATAA as to CVD patients (*p* ≤ 0.0001, Figure 6A–E). However, we can observe that the aortic diameter range affects the level of statistical significance in comparison to the CVD group. Interestingly, TGFB1 expression was not significantly increased in the serum of ATAA patients when compared to the CVD patients whose aorta had a normal diameter (*p* = 0.0640, Figure 6F).

## 4. Discussion

The likelihood of catastrophic aortic complications, such as aortic dissection and rupture, is linked to ascending thoracic aortic aneurysms (ATAA). Guidelines advise preventative surgical replacement of an aneurysmal aorta depending on the size threshold to reduce the likelihood of such complications [16]. However, the majority of aortic aneurysm patients are asymptomatic; hence, early diagnosis of an aneurysm may be challenging. To date, very few studies have specifically focused on the potential value of a blood marker-based approach for clinicians to screen whether patients have ATAA. Previous studies have mainly focused on the identification of biomarkers for aortic aneurysms in general, rather than specifically for the ATAA [21,22]. Furthermore, the majority of previous studies have utilized discovery-based proteomic approaches, such as 2D-gel electrophoresis and mass spectrometry, which have limitations in terms of sensitivity and specificity [23]. In contrast, our study utilized a targeted-proteomic approach, which allowed for the analysis of a specific set of potential biomarkers in a highly sensitive and specific manner.

To the best of our knowledge, this is the first targeted proteomics study investigating the impact of the ascending aorta diameter on protein expression in the serum of ATAA patients, using a targeted proteomic approach. The use of this approach may explain why we were able to identify significant biomarkers for the diagnosis of ATAA, whereas previous studies may have missed these biomarkers due to limitations in their methodology. Overall, our findings suggest that targeted proteomic analysis may be a promising approach for the identification of biomarkers for specific diseases, and further studies using this approach are warranted. The results of our analyses have introduced 6 proteins (CCL5, HBD1, ICAM1, IL8, TNFα, TGFB1) that were differentially expressed in the serum among ATAA patients and control subjects. Our results show that CCL5, HBD1 and ICAM1 help predict ATAA in a time- and cost-effective manner over ultrasound-based screening of ATAA. These biomarkers may assist in the diagnosis and follow-up of patients at risk of developing ATAA. This retrospective study is encouraging and suggests that further study of proteomics is needed to investigate the role of these biomarkers in the pathogenesis of ATAA.

Accurately defining normal size is necessary for clinicians, as increasing aortic size is associated with risk of dissection, rupture and death. Moreover, aortic diameter is used to guide the decision for surgical intervention [16]. In our study, ATAA was defined as an ascending thoracic aorta with a diameter ≥ 4.0 cm. Diagnosis was assessed by echocardiography and confirmed by angio-computed tomography scans. Patients were categorized into three groups depending on aorta diameter: 4.0–4.5 cm (*N* = 23), 4.6–5.0 cm (*N* = 20), >5.0 (*N* = 9) based on the current practice guidelines, as long as recent analyses [1,16,17]. A threshold of 4.0 cm was chosen to maximize the sensitivity of finding an expanded aorta in the screening context [18,19]. While we did observe consistent elevations of these markers (CCL5, HBD1, ICAM1, IL8, TNFα, TGFB1) across all three groups, we did not observe statistically significant intergroup differences for some of these markers (HBD1, ICAM1, IL8). This may suggest that these markers may not be useful in distinguishing between different stages of ATAA progression, and may be more indicative of the presence of ATAA in general.

Until one of the dramatic complications (rupture, intramural hematoma, or dissection) occurs, ATAA is typically a silent disease. To monitor its progress, early diagnosis of ATAA is crucial; hence, improved techniques are required in order to diagnose ATAA before, not after, aortic dissection or rupture happen. Targeted screening requires the development of biomarkers to forecast the patient’s likelihood of having an ATAA in light of the particular difficulties associated with ATAA. However, serum markers are currently unreliable; therefore, imaging is still the primary method for diagnosing and monitoring ATAA, selecting patients who are candidates for repair based on size and symptoms. Until now, routine laboratory tests have not managed to truly assist in the evaluation of ATAA in clinical practice. Increased levels of D-dimer, a degradative product of fibrin, correspond to an increase in the coagulation procedure and secondary hyperfibrinolysis. Therefore, high levels of D-dimer in patients with aneurysm could be explained by the fact that circulating D-dimer in these individuals originates from intraluminal thrombus fibrinolysis [9]. A very recent study has validated the diagnostic efficacy of D-dimer for the prediction of abdominal aortic aneurysm (AAA) in patients with peripheral artery disease [24]. Based on this evidence, we decided to investigate its role as a biomarker for ATAA. However, in our study, we showed that D-dimer cannot be used for the accurate diagnosis of ATAA, since D-dimer showed the least favorable overall diagnostic performance (AUC = 0.64). It is well-known that D-dimer is a sensitive biomarker for thromboembolic events, but it lacks specificity. Conditions other than venous thrombosis can raise blood levels of it, such as renal failure and sepsis [25]. This could explain why for 46.2% of ATAA patients and for 30% of control subjects, D-dimer levels were documented > 0.5 mg/L, even though no pulmonary thromboembolism was found for these individuals. On this note, we should clarify that since D-dimer is incredibly sensitive, we strongly advise using it frequently in emergency rooms for the identification of the development of thrombi in the aortic lumen.

The 12 proteins that were measured in patients’ serum were carefully selected based on their known involvement in various cellular pathways associated with ATAA, such as inflammation, immune cell activation, extracellular matrix degradation, smooth muscle cell proliferation and differentiation, and endothelial dysfunction. To identify candidate biomarkers, we searched several electronic databases including PubMed, Web of Science, and Scopus, using relevant keywords such as “aortic aneurysm”, “proteomics”, “biomarkers”, and “cytokines”. We also consulted with experts in the field to identify additional potential candidates. Although the molecular biological characteristics of aortic aneurysms vary depending on the region of the aorta affected and the risk profile of the patient, a major pathogenic hallmark is inflammation. Inflammation contributes to the occurrence and pathophysiology of the aortic disease through local chronic inflammations causing aortic medial degeneration [16]. Previous studies have reported that inflammation is a key element in AAA, while there is little evidence supporting the inflammatory role in TAA. However, different cells of the immune system (ex. T cells, macrophages) have been observed in human TAA, and several relative inflammatory pathways are reported to be activated [26,27].

Matrix metalloproteinases (MMPs) compose a broad family of calcium-dependent zinc-containing endopeptidases, which can degrade extracellular matrix proteins and have been discovered to have a substantial role in the pathophysiology of aneurysms; they have been recommended as serum markers. Patients with ATAA, AAA and Marfan syndrome have shown elevated MMP-9 production and activity in their aortas [28]. MMPs expression may vary in TAA patients depending on the aneurysm’s location (ascending vs. descending thoracic aorta), etiology (atherosclerotic vs. non-atherosclerotic), size (with increasing values with larger aneurysms), and growth rate (parts of aneurysms with the faster growth, such as anterior vs. posterior wall) [15]. For our targeted proteomic approach, we investigated the levels of MMP-9, but no significant induction of its expression was found in the serum of ATAA patients in comparison to the control subjects (*p* = 0.0861), as well as the CVD patients group (*p* = 0.1587). Even though data regarding the role of MMPs in TAA are continuously expanding, based on our results, MMP9 does not seem useful as a biomarker for ATAA.

Transforming growth factor β (TGF-β) is a protein that in a wide variety of biologic systems is essential in the regulation of cell proliferation, differentiation, and extracellular matrix production [29]. Our hypothesis is that it could be implicated in the pathogenesis of ATAA through its effects on smooth muscle cell apoptosis and matrix remodeling. TGFB has been demonstrated to play a key role in the maturation and function of VSMCs, as well as in aortic development. Evidence suggests that TGFB is protective early in the formation of the aorta, but detrimental at later stages [30]. Interestingly, it has been shown that TGFB signaling could have dimorphic effects on ATAA development. A study on a mouse model of Marfan syndrome showed that Ltbp-3 deficiency was linked to reduced TGFB signaling, less aortic damage and increased survival, suggesting that TGFB signaling is involved in the progression of the aortic disease [31]. In the aortic tissue of patients with familial ATAA, Marfan syndrome, or patients who have mutations in TGFB2, TGFB3, TGFBR or SMAD3, increased TGFB signaling was reported [28]. The underlying mechanisms that could explain this paradoxical rise still remain unknown. Evidence suggests that a lack of or deregulation in canonical TGFB signaling results in a compensatory increase in TGFB production, which, via a non-canonical pathway, can lead to aortic damage and disease development [32]. In our study, the initial results regarding the use of TGFB1 protein as a potential biomarker for the diagnosis of ATAA were promising, since the levels of TGFB1 were increased in the serum of ATAA patients in a significant manner in comparison to the serum of control subjects (*p* = 0.0013). In addition, the diagnostic accuracy of this test was meaningful, with the AUC being greater than 0.5 (AUC = 0.76). However, when we performed the analysis in comparison to the patients with CVD history and normal aortic diameter, no statistical significance was observed (*p* = 0.0640). Thus, we can suggest that TGB1 could be a potential valuable biomarker for CVD, but it cannot perform as a specific clinical test for ATAA diagnosis.

Early experiments with patients with AAA have reported elevated circulating levels of inflammatory cytokines, such as interleukin 1 (IL-1α) and 6 and tumor necrosis factor α (TNFα) [15]. IL-1α is a proinflammatory cytokine that could play a key role in the pathogenesis of ATAA by promoting inflammation and matrix degradation [33]. In our targeted proteomic approach, we aimed to investigate the potential role of IL-1α as a biomarker for AΤAA. However, we did not observe a significant induction of its expression in the serum of ATAA patients compared to the control group (*p* > 0.9999). We also compared the levels of IL-1α between the ATAA group and a group of patients with cardiovascular disease, but no significant difference was detected (*p* > 0.9999). These findings suggest that IL-1α may not be a suitable biomarker for the diagnosis or monitoring of ATAA. TNFα is a proinflammatory cytokine that can promote inflammation, apoptosis, and matrix degradation [34]. A recent study in a Chinese Han population showed that the c.857C/T polymorphism of TNFα promoter was associated with disease progression susceptibility and the CC genotype was associated with increased TNFα expression in patients with thoracic aortic dissection [35]. Our study showed that TNFα expression was significantly increased in ATAA patients when compared to baseline group (*p* < 0.0001), as well as the CVD group (*p* < 0.0001). However, we should mention that its overall diagnostic performance was not exquisite, with an AUC = 0.69. In addition, when compared to the CVD group, significance was achieved only in higher ranges of pathological aortic diameters (>4.6 cm).

Interleukin 8 (IL8) is another proinflammatory cytokine that is involved in the recruitment and activation of neutrophils and other immune cells, which can contribute to the development of ATAA [36]. The main biological function of the chemokine IL8 involves neutrophils and leukocytes chemotaxis and degranulation, and it is reported to induce the cellular expression of intercellular adhesion molecule-1 (ICAM1) [37]. ICAM1, an immunoglobulin-like molecule, is a cell adhesion molecule that is involved in leukocyte recruitment and endothelial cell activation, which can promote inflammation and tissue damage in the aorta [38]. We clearly documented that both IL8 and ICAM1 serum expression was significantly elevated in ATAA patients, in comparison to the control group (*p* < 0.0001). Interestingly, this increase was still significant when compared to the serum of patients with history of cardiovascular disease (CVD group, *p* < 0.0001). In our study, we also evaluated the diagnostic accuracy of two potential biomarkers, IL8 and ICAM1, for the diagnosis of ascending thoracic aortic aneurysm (ATAA). While ICAM1 showed good diagnostic performance with an AUC of 0.83, IL8 showed a fair performance with an AUC of 0.70. This lower performance of IL8 resulted in its exclusion from the potential biomarkers for ATAA diagnosis. However, given the complex interplay of chemokines in promoting inflammatory cell infiltration in aortic aneurysms, we recognize the relevance of IL8 in ATAA pathogenesis and its potential for use in future studies with a larger sample size. Further studies are needed to better understand the role of IL8 and other chemokines in the pathogenesis and early diagnosis of ATAA. Given the variety of chemokines upregulated within aortic aneurysms, it seems likely that IL8 and ICAM1 could interact to promote inflammatory cell infiltration.

A variety of chemokines have been shown to be increased in animal models of aortic aneurysm and in human aortic aneurysm biopsies [39]. C-C motif chemokine ligand 5 (CCL5) and C-X-C motif chemokine 11 (CXCL11) are chemokines that can attract and activate immune cells, leading to inflammation and tissue damage in the aorta [40,41]. Our investigation of CXCL11 levels in the serum of ATAA patients did not reveal a significant induction of its expression compared to the control group (*p* = 0.2481) or the group of patients with cardiovascular disease (*p* = 0.1474). CCL5 is one of the chemokines that is most consequently upregulated in AAA samples [42]. CCL5 is a potent chemoattractant of macrophages, T cells, and dendritic cells, which express chemokine C-C motif receptors (CCR) 1, 3, or 5, which have also been reported to be upregulated in human AAA [37]. This is the first study to assess the expression of CCL5 in the serum of ATAA patients, and our analysis seemed to have identified a potent novel target in the research into effective ATAA biomarkers. The major finding is that CCL5 levels in patients with ATAA were significantly higher than those in participants who did not suffer from ATAA, and their aorta had a normal diameter (*p* < 0.0001). Comparison with patients with CVD revealed that CCL5 is highly specific for the thoracic aortic disease (*p* < 0.0001), and the overall diagnostic performance was considered excellent with AUC = 0.84. It is our suggestion that CCL5 be implemented into clinical practice as a screening tool for ATAA.

In addition, based on the consistent presence of immune-inflammatory cells in tissues of the thoracic aorta, a crucial role for both innate and adaptive immune cells might exist in ATAA evolution. Small cationic peptides known as human defensins are major components of our immune system. Defensins are naturally expressed in a variety of mucosa and epithelial cells in humans; they display potent microcidal properties and they can be activated in response to inflammatory and infectious stimuli. The most important antimicrobial peptide in human epithelia, human beta-defensin 1 (HBD1), is constitutively expressed in the majority of tissues and has an immunomodulatory effect by recruiting immune cells [43]. Our study showed that HBD1 expression was significantly upregulated in the serum of ATAA patients (*p* < 0.0001) and it had an excellent diagnostic accuracy (AUC: 0.83). To our knowledge, this is the first study to reveal a potential functional activity of HBD1 in the development of ATAA. Our hypothesis is that HBD1 secretion could suggest long-term innate immune system activation and subsequent development of a low-grade chronic inflammatory reaction (an ongoing cytokine-induced acute phase response) that elicits disease instead of repair, resulting in chronic inflammation and in the development of ATAA (Figure 7). Further work and in vivo studies on this interesting protein are required to fully understand its functional repertoire and to realize its therapeutic potential.

Our study has a number of limitations. First, this is a retrospective study of living and un-operated subjects. It is essential to conduct a large-scale, multicenter study to further verify the findings. In addition, we should recognize that aortic size differs between individuals depending on body size, age, and sex; therefore, a more elaborate sorting of patients could give further input to the diagnosis of ATAA. For instance, the study population was relatively older, with a mean age of 72 years in both the case groups and the control group. Future studies in younger populations would be beneficial to better understand the role of these biomarkers in the pathogenesis and early diagnosis of ascending thoracic aortic aneurysm. Moreover, we should take into consideration that the group whose aortic diameter was >5.0 cm had a smaller number of participants (*N* = 9) in comparison to the other groups of patients. We should also state that plasma inflammatory biomarkers cannot easily reflect the local inflammatory milieu in the aortic wall. Lastly, since no follow-up procedure was performed, our study cannot directly determine whether an increase in the expression of these makers could contribute to greater risk of acute aortic syndromes and mortality in ATAA patients. Nevertheless, we have uncovered significant candidate proteins that could be used as biomarkers for diagnosis of ascending thoracic aortic disease in a population of patients who are typically at higher risk due to their age.

It is important to note that the biomarkers studied in this research were evaluated for their potential diagnostic value in detecting the presence of ATAA, not for predicting prognosis or replacing imaging tests. The results suggest that these biomarkers, both in combination and individually, may have potential as non-invasive diagnostic tools to aid in the early detection of ATAA. While further validation is needed, these potential biomarkers could provide an easy and sensitive tool for early detection and monitoring of ATAA, which is crucial for timely intervention and management. However, it is important to note that current imaging tests, such as echocardiography and CT scanning, remain the gold standard for the diagnosis and monitoring of ATAA.

## 5. Conclusions

Early detection of people at risk of harboring a clinically silent ATAA is the most efficient way to avoid aortic-related mortality. We have presented a novel comparison of targeted proteomic profiles from serum of subjects with normal human ascending thoracic aorta and of patients with ascending thoracic aneurysmal growth. We suggest performing detection of the serum levels of CCL5, HBD1, and ICAM1 as a simple and useful method for identifying the thoracic aortic disease. Even though there is not enough evidence on inflammatory disease of the thoracic aorta and the mechanism for pathogenesis of ascending aortic aneurysm is still not completely understood, we suggest a pathway that involves long-term innate immune system activation and subsequent development of a low-grade chronic inflammatory reaction for the evolution of ATAA. Understanding the mechanisms and the pathophysiology of ATAA could provide opportunities for developing standardized serological screening tools for the early prognosis of this indolent but virulent disease.

## Figures and Tables

**Figure 1 biomedicines-11-01273-f001:**
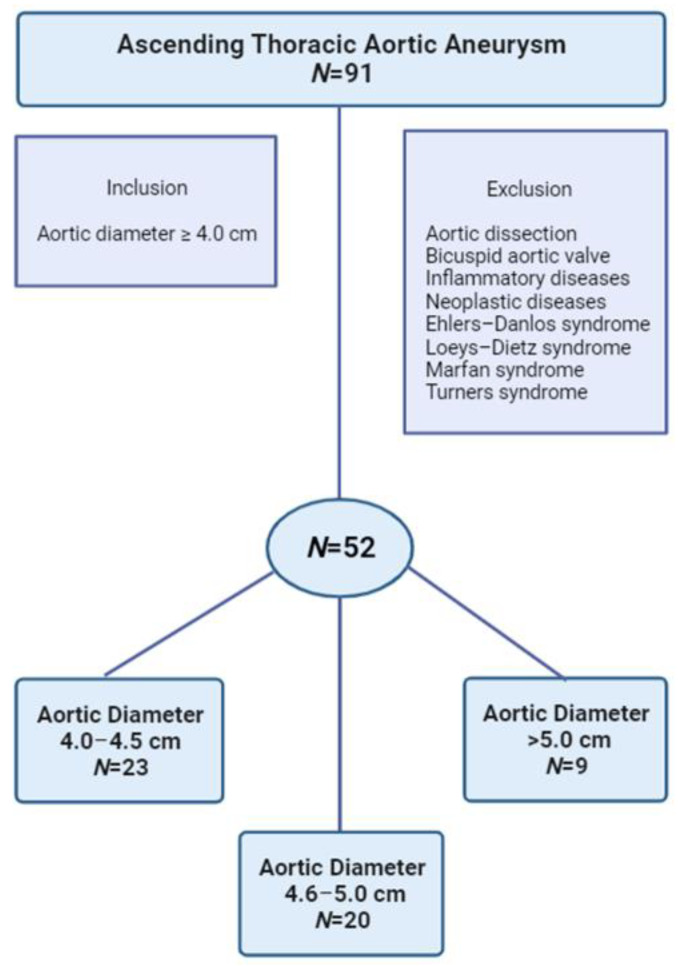
Flowchart for study enrollment.

**Figure 2 biomedicines-11-01273-f002:**
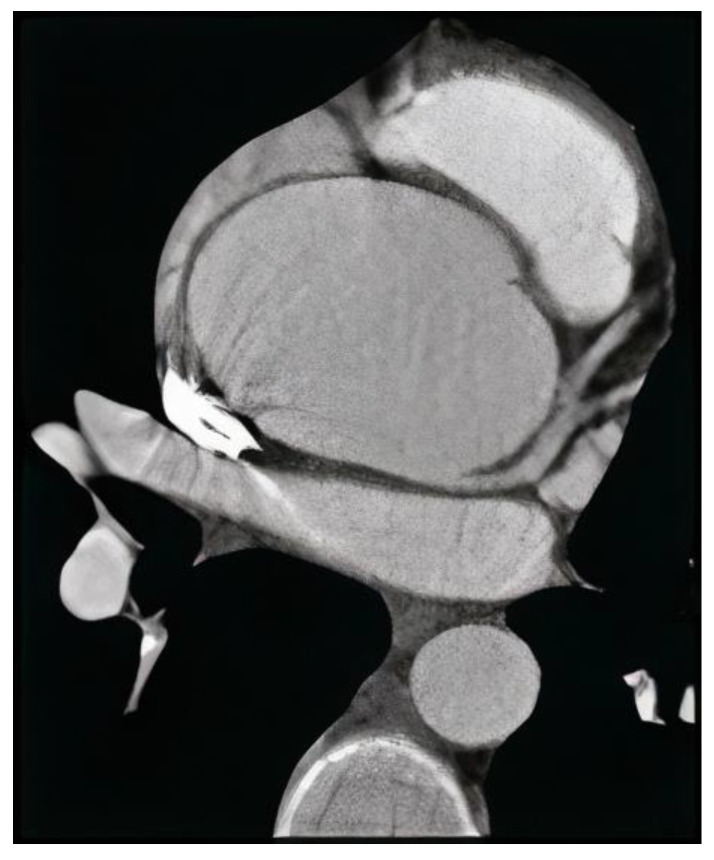
Angio-computed tomography of thoracic aortic aneurysm with electrocardiographic gating.

**Figure 3 biomedicines-11-01273-f003:**
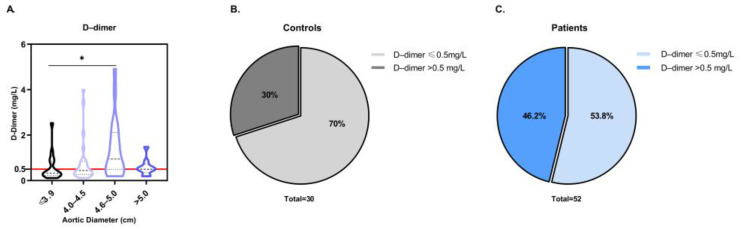
Evaluation of D-dimer in ATAA. (**A**) Violin plot of D-dimer levels in peripheral blood samples of ATAA patients depending on the aortic diameter. *p*-values annotated on the figure. * *p* < 0.05. (**B**) Pie chart illustrating the percentage of control subjects with normal or positive D-dimer. (**C**) Pie chart illustrating the percentage of ATAA patients with normal or positive D-dimer. Positive D-dimer was considered when >0.5 mg/L.

**Figure 4 biomedicines-11-01273-f004:**
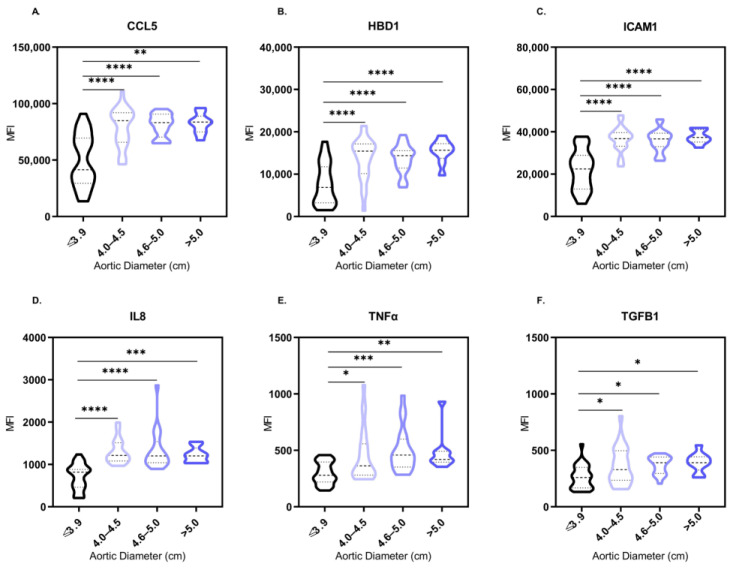
Statistically significant differences of serum MFI levels of CCL5 (**A**), HBD1 (**B**), ICAM1 (**C**), IL8 (**D**), TNFα (**E**) and TGFB1 (**F**) between control subjects and ATAA patients of different aortic diameter range. Distributions expressed in the form of violin plots. Comparisons of proteins’ MFI levels in serum that did not achieve statistical significance (*p* < 0.05) are not illustrated. * *p* < 0.05, ** *p* < 0.01, *** *p* < 0.001, **** *p* < 0.0001. Significance was evaluated using Dunn’s multiple comparison test.

**Figure 5 biomedicines-11-01273-f005:**
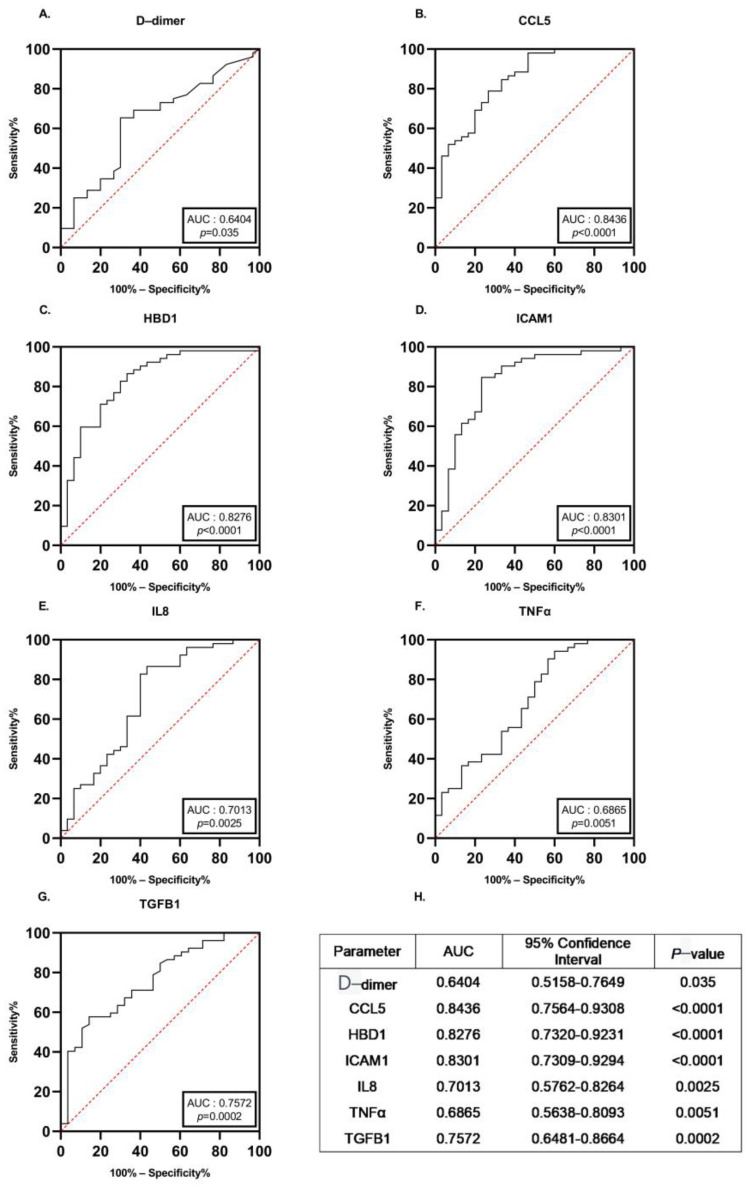
Receiver-operating characteristics curves for all ATAA patients versus all control subjects for each of the potential biomarkers. Receiver operator characteristic curve of D-dimer (**A**), CCL5 (**B**), HBD1 (**C**), ICAM1 (**D**), IL8 (**E**), TNFα (**F**) and TGFB1 (**G**) was plotted in ATAA patients. Diagnostic efficacy of these markers was determined. (**H**) Detailed information of ROC curves. AUC: area under the curve.

**Figure 6 biomedicines-11-01273-f006:**
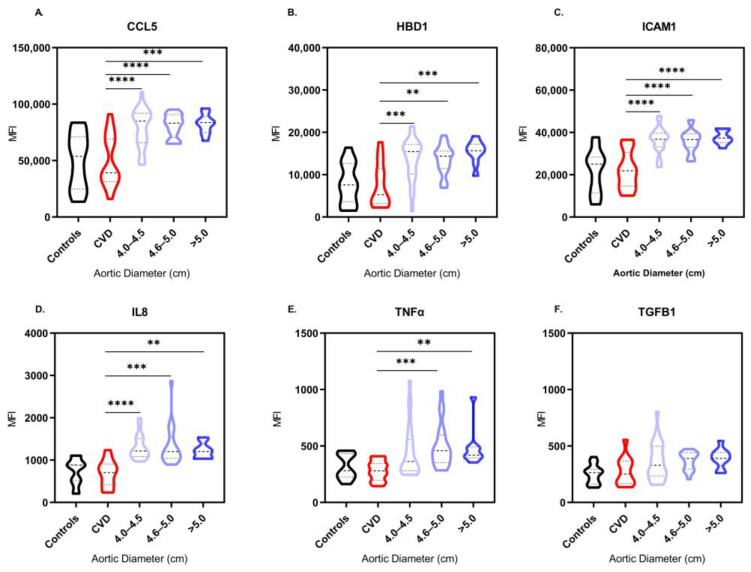
Statistically significant differences of serum MFI levels of CCL5 (**A**), HBD1 (**B**), ICAM1 (**C**), IL8 (**D**), TNFα (**E**) and TGFB1 (**F**) between patients suffering from other cardiovascular diseases (CVD) and ATAA patients of different aortic diameter range. Distributions expressed in the form of violin plots. Comparisons of proteins’ MFI levels in serum that did not achieve statistical significance (*p* < 0.05) are not illustrated. ** *p* < 0.01, *** *p* < 0.001, **** *p* < 0.0001. Significance was evaluated using Dunn’s multiple comparison test.

**Figure 7 biomedicines-11-01273-f007:**
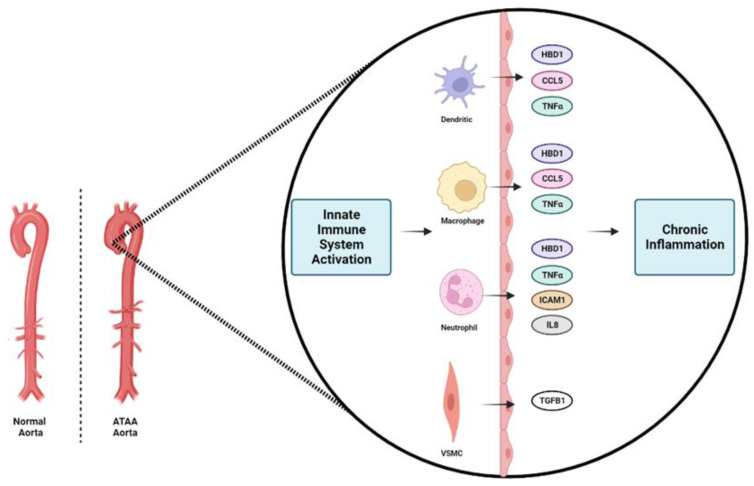
Schematic of our proposed model for the ascending thoracic aortic aneurysm (ATAA) formation. We suggest that long-term innate immune system activation through the release of the investigated proteins (CCL5, HBD1, ICAM1, IL8, TNFα, TGFB1) from neutrophils, dendritic cells, macrophages and vascular smooth muscle cells (VSMCs) occurs in ATAA. This activation results in a low-grade chronic inflammatory reaction that elicits disease instead of repair, resulting in chronic inflammation and in the development of ATAA.

**Table 1 biomedicines-11-01273-t001:** Demographic characteristics of the patients included in our study. Continuous variables are expressed as mean ± standard deviation.

Characteristics	Controls	Patients	*p*-Value
Gender (f:m)	13:17	22:30	0.2048
Age (years)	71.97 ± 11.28	71.27 ± 11.56	0.8952
Height (cm)	165.40 ± 11.09	168.50 ± 8.26	0.3204
Weight (kg)	80.23 ± 13.97	15.60 ± 15.60	0.8951
BMI (kg/m^2^)	29.47 ± 4.56	28.52 ± 4.80	0.3789
BSA (m^2^)	1.85 ± 0.18	1.92 ± 0.18	0.1756
Systolic Blood Pressure (mmHg)	127 ± 11	129 ± 12	0.4243
Diastolic Blood Pressure (mmHg)	69 ± 10	71 ± 10	0.3572
Heart Rate (bpm)	77 ± 13	75.75 ± 11.82	0.7066
Smokers ^a^	11 (36.67%)	25 (48.08%)	0.3616
Hypertension	19 (63.33%)	40 (76.92%)	0.2100
Hyperlipidemia	8 (36.67%)	19 (36.54%)	0.4660
Coronary artery disease	3 (6.67%)	6 (11.54%)	>0.9999
Ascending aorta diameter (cm)	3.39 ± 0.36	4.65 ± 0.32	<0.0001
Aortic arch diameter (cm)	2.79 ± 0.74	2.93 ± 0.58	0.2201
Descending aorta diameter (cm)	2.50 ± 0.45	2.69 ± 0.33	0.0888

^a^ Includes current or former smokers. Continuous variables are presented as mean ± standard deviation and compared using *t*-tests, categorical variables are presented as n (%) and compared using chi-square tests. Abbreviations: bpm, beats per minute; kg, kilograms; cm, centimeters; mmHg, millimeters of mercury.

**Table 2 biomedicines-11-01273-t002:** Glycemic and lipidemic profile of the patients included in our study. Continuous variables are expressed as mean ± standard deviation.

Characteristics	Controls	Patients	*p*-Value
Glucose (mg/dL)	133.80 ± 41.49	124.70 ± 39.82	0.248
Total Cholesterol (mg/dL)	163.09 ± 33.37	157.26 ± 20.84	0.178
HDL-C (mg/dL)	49.66 ± 16.48	47.26 ± 12.18	0.6914
LDL-C (mg/dL)	113.30 ± 36.09	104.40 ± 34.37	0.2616
Triglycerides (mg/dL)	111.50 ± 40.42	125.50 ± 66.63	0.676
Hemoglobin (g/dL)	13.25 ± 1.686	13.84 ± 3.701	0.4215
WBC (cells/mm^3^ × 1000)	8.71 ± 2.85	7.71 ± 1.9	0.1127
Creatinine (mg/dL)	1.00 ± 0.35	1.01 ± 0.35	0.942
SGOT (IU/L)	21.30 ± 7.54	22.75 ± 17.42	0.1322
Urea (mg/dL)	56.09 ± 33.37	43.26 ± 20.84	0.0789
Uric acid (mg/dL)	6.16 ± 1.81	5.75 ± 1.90	0.4437
SGPT (IU/L)	18.88 ± 11.96	19.66 ± 17.24	0.769
γGT (IU/L)	22.96 ± 19.59	29.44 ± 48.93	0.8767
ALP (U/L)	73.37 ± 24.49	74.37 ± 38.50	0.718
Blood amylase (U/mL)	54.56 ± 14.21	69.26 ± 30.72	0.1363
Potassium (mEq/L)	4.51 ± 0.79	4.25 ± 0.41	0.0699
Sodium (mEq/L)	137.60 ± 6.07	140.30 ± 2.6	0.1285
N-terminal prohormone of brain natriuretic peptide (NT-proBNP) (pg/dL)	347.0 ± 252.2	425.5 ± 284.5	0.5806
CRP (mg/dL)	1.71 ± 1.66	3.74 ± 7.37	0.736
Troponine T (ng/mL)	0.06 ± 0.09	0.05 ± 0.13	0.0724

## Data Availability

The data presented in this study are available on request from the corresponding author. The data are not publicly available due to the authors’ wish to maintain control over the use and dissemination of the data to ensure their proper attribution and to prevent potential misuse.

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
