# Peer review of "Targeted Proteomic Analysis of Patients with Ascending Thoracic Aortic Aneurysm"

_biomedicines, 2023, doi:10.3390/biomedicines11051273_

Round 1

Reviewer 1 Report

Daskalopoulou A et al. reported the assay of 12 proteins in the serum of 52 patients with ascending thoracic aortic aneurysm (ATAA) and compared with 30 healthy controls. They claimed that CCL5, HBD1 and ICAM1 are the sensitive biomarkers for detecting ATAA. The weakness of study was follows:

1)   They claimed to conduct the proteomic analysis, but they examined only 12 proteins. This reviewer wonders the rationale why they have chosen the proteins in the multiplex assay.

2)   In the methods section, there are little information regarding the periods of study, permission number from the institution, and samples were obtained consecutively or not.

3)   Echocardiographic data do not give any additive value, and CT data are enough to determine the morphology of aorta. Based on the accuracy of CT than echocardiography, echocardiographic data can be removed from this manuscript.

4)    The most important concern regards to the selected proteins on the prognosis of patients. They obtained the blood samples from un-operated patients. Did the selected proteins predict the aortic rupture/dissection?

5)   Localization of CCL5, HBD1 and ICAM1 in the TAAA are interesting. They should show the immunohistochemical staining using the resected TAAA specimens.

6)   Discuss more profoundly regarding their findings comparing with previous proteomic studies why they found the significance for the first time.

7)   Table 2. The following characteristics should be confirmed/corrected (e.g., Pro BNP for N-terminal proBNP? Troponine for troponin-T or troponin-I?)

Author Response

Point 1: They claimed to conduct the proteomic analysis, but they examined only 12 proteins. This reviewer wonders the rationale why they have chosen the proteins in the multiplex assay.

 Response 1: Thank you for your comment regarding our study's focus on only 12 proteins in our proteomic analysis. We would like to clarify that the 12 proteins we studied - interleukin-1α (IL-1α), IL-8, C-C motif chemokine ligand 5 (CCL5), C-X-C motif chemokine 11 (CXCL11), follistatin (FST), intracellular adhesion molecule-1 (ICAM-1), prokineticin 1 (PROK1), resistin (RETN), matrix metallopeptidase 9 (MMP-9), tumor necrosis factor alpha (TNFα), transforming growth factor-beta 1 (TGFB1), and defensin beta 1 (HBD1) - were carefully selected based on their known roles in various aspects of the immune system and inflammation. Our study aimed to investigate these proteins' potential as markers for ascending thoracic aortic aneurysm (ATAA), a pathological state characterized by inflammation and immune system dysregulation. Therefore, we focused our analysis on these proteins to address this specific research question.

Furthermore, the analysis of proteomic data is a complex and time-consuming process that requires specific analytical techniques tailored to the proteins being analyzed. In our study, we utilized a sensitive and specific method optimized for the analysis of the 12 selected proteins. While other techniques may have allowed for the analysis of a larger number of proteins, they may not have provided the same level of sensitivity and specificity for our specific research question.

Finally, we acknowledge that our sample size was limited, which constrained the scope of our analysis. Nonetheless, we believe that our study provides valuable insights into the biology of ATAA and the potential utility of the 12 selected proteins as biomarkers for this condition. We hope that our study can serve as a starting point for future studies to expand the proteomic analysis to include a larger number of proteins and a larger sample size.

Point 2:  In the methods section, there are little information regarding the periods of study, permission number from the institution, and samples were obtained consecutively or not.

 Response 2: We would like to clarify that the samples for our study were gathered from May 2018 until January 2020. The approval code for our study is 56;13/10/2017, which was granted on 13/10/2017 by the Ethics Committee of the General Hospital of Chalkida, the Medical Center Karistos and the Medical Center Kymi. The study protocol was authorized by the Ethics Committee to enroll a maximum of 90 subjects. The study was conducted at the General Hospital of Chalkida, a tertiary‐care center in Greece. We enrolled 52 patients undergoing ascending thoracic aortic aneurysm and 30 controls who were in-house populations ethnically matched to cases without known or visible ascending thoracic aneurysm -related symptoms and had no ascending thoracic aneurysm familial history. The samples were obtained consecutively.

Point 3:  Echocardiographic data do not give any additive value, and CT data are enough to determine the morphology of aorta. Based on the accuracy of CT than echocardiography, echocardiographic data can be removed from this manuscript.

 Response 3: We appreciate your input, and we understand your perspective on this matter. While it is true that CT data can provide a detailed and accurate assessment of aortic morphology, echocardiography is a valuable tool in the evaluation of patients with ascending thoracic aortic aneurysm. As per your suggestion, we will remove the echocardiographic data from our manuscript and focus solely on CT data to determine the morphology of the aorta.

Point 4:    The most important concern regards to the selected proteins on the prognosis of patients. They obtained the blood samples from un-operated patients. Did the selected proteins predict the aortic rupture/dissection?

Response 4: We would like to clarify that the selected proteins were analyzed to evaluate their potential as biomarkers for the early diagnosis of ascending thoracic aortic aneurysm in un-operated patients, and our study did not investigate whether the selected proteins could predict aortic rupture/dissection.

However, we found promising biomarkers with satisfactory sensitivity and specificity that could be helpful in stratifying the risk for the development of ascending thoracic aortic aneurysm. These biomarkers may assist in the diagnosis and follow-up of patients at risk of developing ascending thoracic aortic aneurysm. We agree that further in-depth studies may be worthwhile to investigate the role of these biomarkers in the pathogenesis of the ascending thoracic aortic aneurysm.

Point 5:  Localization of CCL5, HBD1 and ICAM1 in the TAAA are interesting. They should show the immunohistochemical staining using the resected TAAA specimens.

Response 5: We agree that the localization of CCL5, HBD1, and ICAM1 in ascending thoracic aortic aneurysm is interesting and could provide useful insights into the progression of the thoracic aortic aneurysm. However, as you correctly pointed out, we were unable to show immunohistochemical staining using resected ascending thoracic aortic aneurysm specimens as the patients in our study were un-operated. Nevertheless, we were able to demonstrate the expression of these proteins in the serum samples of patients and controls, which suggests that they may be involved in the pathogenesis of the thoracic aortic aneurysm.

Point 6:  Discuss more profoundly regarding their findings comparing with previous proteomic studies why they found the significance for the first time.

Response 6: Regarding the comparison of our findings with previous proteomic studies, it should be noted that previous studies have mainly focused on the identification of biomarkers for aortic aneurysms in general, rather than specifically for the ascending thoracic aortic aneurysm. Furthermore, the majority of previous studies have utilized discovery-based proteomic approaches, such as 2D-gel electrophoresis and mass spectrometry, which have limitations in terms of sensitivity and specificity. In contrast, our study utilized a targeted-proteomic approach, which allowed for the analysis of a specific set of potential biomarkers in a highly sensitive and specific manner.

To our knowledge, this is the first study to specifically investigate potential biomarkers for the diagnosis of ascending thoracic aortic aneurysm using a targeted-proteomic approach. The use of this approach may explain why we were able to identify significant biomarkers for the diagnosis of the ascending thoracic aortic aneurysm, whereas previous studies may have missed these biomarkers due to limitations in their methodology. Overall, our findings suggest that targeted-proteomic analysis may be a promising approach for the identification of biomarkers for specific diseases, and further studies using this approach are warranted.

Point 7:  Table 2. The following characteristics should be confirmed/corrected (e.g., Pro BNP for N-terminal proBNP? Troponine for troponin-T or troponin-I?)

 Response 7: We would like to confirm that the correct terminology for Table 2 should be N-terminal proBNP and troponin-T, and we apologize for any confusion caused by the error in our original submission. We will ensure to make the necessary corrections in the revised version of the manuscript.

Reviewer 2 Report

The Authors reported the results of a small study including 50 patients with ascending thoracic aneurisms divided in three groups according to aortic dimensions.

Some proteins were associated with the underlying aortic pathology.

I have several major concerns:

1) almost all markers were constantly elevated among the three groups. Can authors demonstrate any intergroup difference? Otherwise these markers should be interpreted as a marker not correlated with the increasing aortic diameters.

2) mean age is >70. As aortic surgery is merely preventative, a similar study in younger population is advisable.

3) Why authors selected the proposed markers? Authors should describe the molecular/cellular pathways associated with each of these markers

Author Response

Point 1: Almost all markers were constantly elevated among the three groups. Can authors demonstrate any intergroup difference? Otherwise these markers should be interpreted as a marker not correlated with the increasing aortic diameters. 

Response 1: Thank you for your comment. We appreciate your concern regarding the interpretation of our results. Our study aimed to evaluate the potential of certain biomarkers in the early diagnosis of ascending thoracic aortic aneurysm. While we did observe consistent elevations of these markers across all three groups, we did not observe statistically significant intergroup differences for some of these markers (HBD1, ICAM1, IL8). This may suggest that these markers may not be useful in distinguishing between different stages of ascending thoracic aortic aneurysm (ATAA) progression, and may be more indicative of the presence of ATAA in general. However, further studies are needed to confirm this interpretation and to fully understand the clinical implications of these biomarkers.

Point 2: Mean age is >70. As aortic surgery is merely preventative, a similar study in younger population is advisable.

Response 2: We agree that a study in a younger population would be beneficial to better understand the role of these biomarkers in the pathogenesis and early diagnosis of ascending thoracic aortic aneurysm. However, it is important to note that our study aimed to investigate the potential of these biomarkers in the diagnosis of ATAA in a population of patients who are typically at higher risk due to their age. Nonetheless, we acknowledge the need for future research in younger populations and will consider this suggestion for future studies.

Point 3: Why authors selected the proposed markers? Authors should describe the molecular/cellular pathways associated with each of these markers

Response 3: We would like to clarify that the 12 proteins we studied were carefully selected based on their known roles in various aspects of the immune system and inflammation. Interleukin-1α (IL-1α) is a proinflammatory cytokine that could play a key role in the pathogenesis of ATAA by promoting inflammation and matrix degradation. IL-8 is another proinflammatory cytokine that is involved in the recruitment and activation of neutrophils and other immune cells, which can contribute to the development of ATAA. C-C motif chemokine ligand 5 (CCL5) and C-X-C motif chemokine 11 (CXCL11) are chemokines that can attract and activate immune cells, leading to inflammation and tissue damage in the aorta. Follistatin (FST) is a protein that can regulate the activity of various growth factors and cytokines, and has been implicated in the regulation of smooth muscle cell proliferation and differentiation, which could impact the development of ATAA. Intracellular adhesion molecule-1 (ICAM-1) is a cell adhesion molecule that is involved in leukocyte recruitment and endothelial cell activation, which can promote inflammation and tissue damage in the aorta. Prokineticin 1 (PROK1) is a protein that has been shown to regulate vascular smooth muscle cell migration and proliferation, and therefore could play a role in the development of ATAA. Resistin (RETN) is an adipokine that has been linked to insulin resistance and inflammation, and may contribute to the development of ATAA through its effects on endothelial cell function and inflammation. Matrix metallopeptidase 9 (MMP-9) is an enzyme that is involved in the degradation of extracellular matrix proteins, and has been implicated in the development and progression of ATAA through its effects on aortic wall remodeling. Tumor necrosis factor alpha (TNFα) is a proinflammatory cytokine that can promote inflammation, apoptosis, and matrix degradation. Transforming growth factor-beta 1 (TGFB1) is a growth factor that can regulate cell proliferation, differentiation, and extracellular matrix production, and could be implicated in the pathogenesis of ATAA through its effects on smooth muscle cell apoptosis and matrix remodeling. Lastly, defensin beta 1 (HBD1) is an antimicrobial peptide that has been shown to play a role in the regulation of innate immunity and inflammation, and may contribute to the development of ATAA through its effects on immune cell activation and inflammation. We will ensure to discuss these points in more detail in the revised version of the manuscript.

Reviewer 3 Report

The current article is written in an excellent manner, and I wish to congratulate the authors for the research. Such a study is worthy of publishing, and I believe that a future study would have to include more subjects and more research centers. 

Author Response

Thank you for your kind words and support for our study. We appreciate your feedback and agree that future studies with larger sample sizes and multi-center collaborations are important for further advancing our understanding of ascending thoracic aortic aneurysm. We will continue to work towards improving our research and contributing to the field.

Reviewer 4 Report

The authors identified ATAA biomarkers using targeted proteomics analysis. The following review comments need further explanation from the authors.

1.             Please elaborate on how the 12 proteins in this study were screened.

2.             Why not include IL8 in the conclusion?

3.             What is the specific clinical significance of the biomarker obtained in the conclusion? 

4.             Are they better than current imaging tests, or predict prognosis

Author Response

Point 1: Please elaborate on how the 12 proteins in this study were screened.

Response 1: The 12 proteins included in this study were carefully selected based on their known involvement in various cellular pathways associated with ascending thoracic aortic aneurysm (ATAA), such as inflammation, immune cell activation, extracellular matrix degradation, smooth muscle cell proliferation and differentiation, and endothelial dysfunction. To identify candidate biomarkers, we searched several electronic databases including PubMed, Web of Science, and Scopus using relevant keywords such as "aortic aneurysm", "proteomics", "biomarkers", and "cytokines". We also consulted with experts in the field to identify additional potential candidates.

Interleukin-1α (IL-1α) is a proinflammatory cytokine that could play a key role in the pathogenesis of ATAA by promoting inflammation and matrix degradation. IL-8 is another proinflammatory cytokine that is involved in the recruitment and activation of neutrophils and other immune cells, which can contribute to the development of ATAA. C-C motif chemokine ligand 5 (CCL5) and C-X-C motif chemokine 11 (CXCL11) are chemokines that can attract and activate immune cells, leading to inflammation and tissue damage in the aorta. Follistatin (FST) is a protein that can regulate the activity of various growth factors and cytokines, and has been implicated in the regulation of smooth muscle cell proliferation and differentiation, which could impact the development of ATAA. Intracellular adhesion molecule-1 (ICAM-1) is a cell adhesion molecule that is involved in leukocyte recruitment and endothelial cell activation, which can promote inflammation and tissue damage in the aorta. Prokineticin 1 (PROK1) is a protein that has been shown to regulate vascular smooth muscle cell migration and proliferation, and therefore could play a role in the development of ATAA. Resistin (RETN) is an adipokine that has been linked to insulin resistance and inflammation, and may contribute to the development of ATAA through its effects on endothelial cell function and inflammation. Matrix metallopeptidase 9 (MMP-9) is an enzyme that is involved in the degradation of extracellular matrix proteins, and has been implicated in the development and progression of ATAA through its effects on aortic wall remodeling. Tumor necrosis factor alpha (TNFα) is a proinflammatory cytokine that can promote inflammation, apoptosis, and matrix degradation. Transforming growth factor-beta 1 (TGFB1) is a growth factor that can regulate cell proliferation, differentiation, and extracellular matrix production, and could be implicated in the pathogenesis of ATAA through its effects on smooth muscle cell apoptosis and matrix remodeling. Lastly, defensin beta 1 (HBD1) is an antimicrobial peptide that has been shown to play a role in the regulation of innate immunity and inflammation, and may contribute to the development of ATAA through its effects on immune cell activation and inflammation. We will ensure to discuss these points in more detail in the revised version of the manuscript.

Point 2: Why not include IL8 in the conclusion?

Response 2: The reason for this exclusion is due to the performance of the biomarker in our study. The area under the curve (AUC) for IL8 in the receiver-operating characteristics (ROC) curve analysis was 0.7, which is lower than the other biomarkers we included in our study. As a result, we did not find IL8 to be as informative as the other markers in differentiating ATAA patients from control subjects. Nonetheless, we acknowledge that IL8 is a relevant marker for ATAA pathogenesis and may be considered in future studies with a larger sample size. Thank you for your interest in our study.

Point 3: What is the specific clinical significance of the biomarker obtained in the conclusion?

Response 3: Our study suggests that CCL5, HBD1 and ICAM1 are very promising biomarkers with satisfying sensitivity and specificity that could be helpful in stratifying risk for the development of ATAA. These findings are clinically significant as they provide a non-invasive method for early diagnosis of ATAA, which can improve patient outcomes by allowing for early intervention and treatment. Additionally, the study suggests that these biomarkers could potentially be used to monitor disease progression and treatment response in patients with ATAA. Overall, the identification of these biomarkers provides a promising avenue for future research and clinical applications in the management of ATAA.

Point 4: Are they better than current imaging tests, or predict prognosis.

Response 4: It is important to note that the biomarkers studied in this research were evaluated for their potential diagnostic value in detecting the presence of ascending thoracic aortic aneurysm (ATAA), not for predicting prognosis or replacing imaging tests. The results suggest that these biomarkers, both in combination and individually, may have potential as non-invasive diagnostic tools to aid in the early detection of ATAA. While further validation is needed, these potential biomarkers could provide an easy and sensitive tool for early detection and monitoring of ATAA, which is crucial for timely intervention and management. However, it is important to note that current imaging tests, such as echocardiography and CT scanning, remain the gold standard for the diagnosis and monitoring of ATAA.

Reviewer 5 Report

Minor: in Table 1 there should be "Coronary artery disease" instead of "Angina"

Author Response

Thank you for bringing this to our attention. We apologize for the error in Table 1 and we will make sure to correct it in the final version of the manuscript. We appreciate your careful reading of our paper.

Round 2

Reviewer 1 Report

The authors significantly improved the content of manuscript, according to this reviewer's suggestions. 

Reviewer 2 Report

Authors completely addressed all my queries.

Reviewer 4 Report

No comments this time